# Photoperiod Induces DNA Methylation Changes in the Melatonin Receptor 1A Gene in Ewes

**DOI:** 10.3390/ani13121917

**Published:** 2023-06-08

**Authors:** Xiaoyun He, Wei Wang, Wei Sun, Mingxing Chu

**Affiliations:** 1Key Laboratory of Animal Genetics and Breeding and Reproduction of Ministry of Agriculture and Rural Affairs, Institute of Animal Science, Chinese Academy of Agricultural Sciences, Beijing 100193, China; hexiaoyun@caas.cn (X.H.); wangw8182@163.com (W.W.); 2College of Animal Science and Technology, Yangzhou University, Yangzhou 225009, China; dkxmsunwei@163.com; 3Joint International Research Laboratory of Agriculture and Agri-Product Safety of Ministry of Education of China, Yangzhou University, Yangzhou 225009, China

**Keywords:** photoperiod, hypothalamus, melatonin receptor 1A, promoter, DNA methylation

## Abstract

**Simple Summary:**

Accurate timing of photoperiod changes is vital for animal reproduction. Previous research has shown that the polymorphisms and expression of melatonin receptor 1A (MTNR1A) play essential roles in changes in the ovine estrus cycle and seasonal reproduction. In this study, we measured the expression of MTNR1A in the ovine hypothalamus after different photoperiod treatments (short photoperiod treatment for 42 days, SP42; long photoperiod treatment for 42 days, LP42; 42 days of short photoperiod followed by 42 days of long photoperiod treatments, SP-LP42). Subsequently, the core promoter and its methylation levels at CpG sites in the *MTNR1A* gene in different photoperiod treatment groups were identified. We found that the core promoter region in *MTNR1A* was located in a 540 bp region upstream of the transcription start site (TSS) and that the difference in the DNA methylation levels at CpG sites or the expression of MTNR1A was significantly varied in different photoperiod groups. These results suggested that the photoperiod induced the DNA methylation of the *MTNR1A* gene, changing its expression. The finding of photoperiod-induced DNA methylation in the *MTNR1A* gene is valuable for the further study of seasonal reproduction in sheep.

**Abstract:**

Research has shown that MTNR1A plays an essential role in the estrus cycle and seasonal reproduction changes in sheep. However, few people have focused on the DNA methylation of MTNR1A by season or photoperiod. In this study, using qPCR and Western blotting, we measured the MTNR1A expression in the hypothalamus of ovariectomized and estradiol-treated (OVX + E2) sheep under different photoperiod treatment conditions. Subsequently, the core promoter of the *MTNR1A* gene was identified, and its methylation level in sheep exposed to different photoperiod treatments was measured by pyrosequencing. The results showed that *MTNR1A* gene expression significantly differed between the short 42-day photoperiod (SP42) and the LP42 or combined SP-LP42 treatment groups (*p* < 0.05). In addition, we determined that the core *MTNR1A* promoter region was within 540 bp upstream of the transcriptional start site (TSS) and that the DNA methylation levels at CpG sites in the SP42 vs. LP42 and SP42 vs. SP-LP42 groups significantly differed. Finally, a significant negative correlation (*p* < 0.001) between gene expression and DNA methylation levels was revealed, suggesting that photoperiod may induce DNA methylation of the *MTNR1A* gene and thus change its expression. The findings provide valuable bases for the further study of seasonal reproduction in sheep.

## 1. Introduction

Animals with seasonal breeding behaviors are sensitive to changes in photoperiods, which guide seasonal reproduction and ensure the survival of the next generation, and most mammals harbor a highly precise mechanism for sensing photoperiods and show marked behavioral changes in response to small photoperiod alterations. In animals that live at temperate latitudes, changes related to photoperiods are temporal signals that initiate changes in hormone secretion and reproductive status [1]. Changes in the photoperiod are translated into physiological signals through nighttime secretion of melatonin (MLT) from the pineal gland. Then, MLT, binding to its receptor (MTNR), plays multiple biological functions, including the regulation of animal sexual behavior [2], early development [3], seasonal reproduction [4], and circadian rhythm [5]. Mammalian MLT receptors can be classified into two subtypes, melatonin receptor 1A (MTNR1A) and MTNR1B, and numerous studies have shown that MLT primarily binds with MTNR1A in the suprachiasmatic nucleus (SCN) and that the hypothalamus plays an important role in animal reproduction [6,7]. A polymorphism in *MTNR1A* gene exon 2 is associated with reproduction in several sheep breeds, including Aragonesa [8], Dorset [9], Awassi [4], and Sarda [10], and the expression of MTNR1A in the hypothalamus is affected by seasonal photoperiod-induced steroid hormone secretion during the estrous cycle [11]. Our previous study revealed that the expression of the *MTNR1A* gene in ovariectomized ewes changed with photoperiod variations [12].

Recently, several studies have indicated that DNA methylation and histone acetylation modification contributes to the regulated expression of genes encoding hypothalamic peptides that control reproductive activities. Tomikawa’s studies showed that estrogen induces the recruitment of estrogen receptor α and histone acetylation at the *Kiss1* promoter region of AVPV and thus enhances chromatin loop formation in the Kiss1 promoter and *Kiss1* gene enhancer, increasing AVPV-specific *Kiss1* gene expression [13]. A positive correlation between *Kiss1* gene expression and histone H3 acetylation of Kiss1 during different estrous stages was also reported in sheep in our previous study [14]. In addition, DNA methylation plays a vital role in the regulation of hypothalamus-related reproductive functions. Several differentially methylated CpG or CpH genes, including *MAX*, *MMP2*, *FGF11*, and *GSK3B*, have been significantly associated with puberty, as determined via a genome-wide DNA methylation analysis of the hypothalamus in sheep [15], goats [16], yak [17], and mice [18]. In the hamster hypothalamus, short photoperiods and winter-like MLT levels inhibited hypothalamic DNA methyltransferase expression and reduced type Ⅲ deiodinase (*dio3*) promoter DNA methylation, which upregulated dio3 expression and induced gonadal regression, while refractoriness to short vernal photoperiods reestablished summer-like methylation of the dio3 promoter, dio3 expression, and reproductive competence [19]. In addition, DNA methylation changes suggest a diapause response in wasps; specifically, the photoperiodic timing in *Nasonia* was induced, and when DNA methyltransferase 1a (Dnmt1a) and Dnmt3 expression was knocked down or DNA methylation was pharmacologically blocked, the photoperiodic diapause response was largely disrupted in these wasps [20].

To date, most studies have focused on the association between the expression of the *MTNR1A* gene or its polymorphisms and ovine seasonal reproduction. Although large-scale “methylome” data indicate that cytosine methylation is prevalent at CpG islands in the promoters of important reproductive genes, such as the *dio3* gene, little research has been conducted on MTNR1A methylation. We speculate that the expression of the MTNR1A gene may also be regulated by DNA methylation based on our previous study [12]. Therefore, in order to explore whether photoperiod changes induce DNA methylation, resulting in significant differences in gene or protein expression, we analyzed the DNA methylation level of the CpG island near the *MTNR1A* gene core promoter region by distinguishing differences in gene expression under different photoperiod conditions. The results may provide new insights into the photoperiod response (seasonal estrus) due to epigenetic modification in sheep.

## 2. Materials and Methods

### 2.1. Animals and Sample Collection

A cohort of nine clinically normal, non-pregnant Sunite ewes (weighing between 35 and 40 kg and aged three years) were selected from Urat Middle Banner, BayanNur City, Inner Mongolia Autonomous Region, China, and subsequently housed at a farm located at Tianjin, China. These ewes were provided with ad libitum access to food and water. The bilateral ovariectomy and estradiol treatments were previously described [12,21]. Briefly, the estradiol (E2) treatment was administered through implants with an inner diameter of 3.35 mm and an outer diameter of 4.65 mm, containing 20 mg crystalline 17β-estradiol (Sigma Chemical Co., St. Louis, MO, USA). The axillary region was the site of implantation for devices intended to generate circulating E2 levels of approximately 3–5 pg/mL. Subsequently, an equivalent number of ewes were allocated to three photoperiod-controlled chambers (SP, an abbreviated photoperiod of 8/16 h light–dark; LP, an extended photoperiod of 8/16h light–dark; and SP that was converted to an LP). Following photoperiodic treatments, the sheep were humanely euthanized on SP day 42 (SP42), LP42, and SP-LP42, and the hypothalamic tissues were promptly extracted from the brain, rinsed with PBS (pH 7.4), flash-frozen in liquid nitrogen, and preserved at −80 °C for subsequent analysis.

### 2.2. qPCR and Western Blotting

The isolation of total RNA from each sample was carried out using a TRIzol reagent (Invitrogen, Carlsbad, CA, USA) followed by the detection of RNA degradation and contamination using 1% agarose gels prior to subsequent analysis. Subsequently, cDNA was synthesized from the RNA samples post-sequencing using a PrimeScript™ RT reagent kit (TaKaRa, Dalian, China). The expression of the MTNR1A gene (GenBank accession no. NM_001009725.1) was measured using specific primers listed in Table 1. qPCR was performed on a Roche LightCycler 480 (Roche Applied Science, Mannheim, Germany) using a TB Green assay kit (TaKaRa, Dalian, China). The qPCR mixture and program have been described [22], and ACTB was used as the reference gene. In addition, the hypothalamus tissues were also lysed with a proteinase inhibitor-containing lysis buffer to isolate total proteins. A 12% SurePAGE gel (GenScript, Nanjing, China) was used to separate equivalent amounts of the isolated protein. After electrophoresis, the separated proteins were transferred onto a PVDF membrane (Pall, Emiliano Zapata, Mexico), which was sealed with a sealing solution (Tiangen, Beijing, China). The blocked membrane was incubated overnight at 4 °C with anti-MTNR1A antibody (1:1000, ab87639, Abcam, Boston, MA, USA), and the GAPDH (1:2000, ab263962, Abcam) was used as the internal reference. After rinsing 3 times with Tris-buffered saline/Tween, the corresponding HRP-labeled sheep anti-goat IgG (1:5000, Proteintech, Chicago, IL, USA) was used to incubate the membranes for 1 h at room temperature. The protein blots were visualized with an enhanced chemiluminescent reagent (Beyotime, Shanghai, China).

### 2.3. Isolation of the 5′-Flanking Region and Luciferase Reporter Vector Construction

A 2000 bp segment including the transcription start site (TSS) of the ovine *MTNR1A* gene (GenBank: NM_001009725.1) was obtained to identify the core promoter, and the purified PCR products were cloned into a pMD18-T vector (TaKaRa, Dalian, China). Then, the sequence of the recombinant vector was confirmed by sequencing. To produce luciferase reporter constructs including MNTR1A promoter fragments of different sizes, we truncated the promoter and subcloned it into a promoterless pGL3-basic vector (Promega, Wisconsin, USA). The recombinant constructs were named pGL3-P1 (−1/−540), pGL3-P2 (−1/−800), pGL3-P3 (−1/−1080), pGL3-P4 (−1/−1560), pGL3-P5 (−1/−1780), and pGL3-P6 (−1/−2000). All these vectors were used to conduct the subsequent experiments.

### 2.4. Cell Culture, Transfection, and Core Promoter Identification via Luciferase Assay

293T cells were maintained in a medium (DMEM, Gibco, Boston, MA, USA) supplemented with 10% fetal bovine serum (FBS, Gibco) and incubated at 37 °C in 5% CO_2_. Before transfection, the 293 cells were plated at 0.5 × 10^5^ cells/well in 24-well plates for culturing overnight. Subsequently, transient transfection was performed with the promoter luciferase reporter constructs using Lipofectamine™ 2000 (Invitrogen, Carlsbad, CA, USA). Each well was transfected at a 19:1 ratio of promoter luciferase reporter plasmids or positive control plasmid pGL3-control vector (Promega, Madison, WI, USA) to an internal control plasmid expressing Renilla luciferase called a pRL-TK vector (Promega, Madison, WI, USA), and the ratio of total plasmid DNA to the Lipofectamine™ 2000 transfection reagent was 1:2.5. The cells were transfected with each recombinant plasmid in triplicate. After 24 h of transfection, the cells were lysed, and firefly and Renilla luciferase activities were analyzed with a Dual-Luciferase^®^ Reporter Assay System (Promega, Madison, WI, USA), while luminescence was determined on a Tecan Infinite^®^ 200 Pro (Tecan Group LTD, Männedorf, Switzerland). Firefly luciferase activity levels were normalized to those of the Renilla luciferase (pRL-TK) in each well, and the observed values were compared with the value of the negative control luciferase vector pGL3-basic.

### 2.5. DNA Isolation and Bisulfite Treatment

Genomic DNA from each sample was extracted using the phenol-chloroform method and then dissolved in ddH_2_O. After extraction, an EpiTect Bisulfite kit (QIAGEN, Dusseldorf, Germany) was used for bisulfite treatment according to the manufacturer’s instructions. Briefly, a 140 μL reaction system was used for base conversion: 1 μg of DNA (20 μL), 85 µL of bisulfite mix, and 35 µL of DNA protection buffer. After chemical conversion, unmethylated cytosine bases were converted into uracil bases, and methylated cytosine bases were protected. The converted DNA was then extracted for subsequent DNA methylation analysis.

### 2.6. Pyrosequencing Analysis

A pyrosequencing protocol was employed to measure the DNA methylation of the *MTNR1A* gene core promoter, specifically methylation on a specific CpG island. Pyrosequencing amplification and sequencing primers were designed by Assay Design Software with PyroMark Assay Design 2.0 (QIAGEN). The primer information and sizes of the fragments produced are shown in Table 2. PCR was performed in a volume of 25 μL according to the PyroMark^®^ PCR kit instructions: 2.5 μL of 10 × PCR CoraLoad Concentrate, 5 μL of 5 × Q-solution, 0.5 μL of each primer (10 μM), 12.5 μL of 2× PyroMark PCR Master Mix, and 50 ng of bisulfite-treated genomic DNA. The PCR amplification conditions were as follows: denaturation at 95 °C for 15 min, then 45 cycles of 94 °C for 30 s, optimal annealing temperature for each specific primer for 30 s, 72 °C for 30 s, and a final hold at 20 °C. Subsequently, the pyrosequencing primer was used for sequencing the PCR products (Table 2). The sequencing reaction was conducted in the PyroMark Q96 system (QIAGEN) with a PyroMark Gold Q96 Reagents Kit (QIAGEN, Dusseldorf, Germany) according to the system manufacturer’s instructions.

### 2.7. Statistical Analysis

Statistical evaluation of the data was performed using SPSS version 22. One-way ANOVA and paired-sample *t*-tests were performed for statistical analysis. Pearson correlation coefficients were calculated to indicate the correlation between DNA methylation and gene expression levels, and the significance was also determined. The data are presented as the mean ± standard error (SE) values of independent determinations. *p* < 0.05 and *p* < 0.01 are considered to be statistically significant and highly significant, respectively.

## 3. Results

### 3.1. Expression Differences of MTNR1A in Different Photoperiod Groups

To determine the expression level of MTNR1A under different photoperiod conditions, mRNA and protein expression levels in the hypothalamus were measured. The expression of MTNR1A mRNA in the SP42 group was significantly higher than that in the SP42 and SP-LP42 groups (*p* < 0.05) (Figure 1A). Western blot and grayscale analyses showed that the protein expression of MTNR1A significantly differed between the SP42 vs. SP-LP42 groups and the SP42 vs. LP42 groups (*p* < 0.05) (Figure 1B,C). The MTNR1A expression trend at the mRNA and protein levels was consistent for different photoperiod treatments. These results indicated that the photoperiod exerted a significant effect on the expression of MTNR1A.

### 3.2. Identification of the Core Promoter in the MTNR1A Gene

To identify the *MTNR1A* gene core promoter, we constructed a series of deletion luciferase reporter constructs and transfected them into HEK293T cells. The luciferase activity of the different-sized fragments was measured using a dual luciferase reporter system. The luciferase activity levels of the luciferase reporter constructs in HEK293T cells were higher than those in the PGL3-basic-expressing cells (Figure 2). The cells expressing pGL3-P1 showed a significant increase in the luciferase activity level compared with those expressing pGL3-basic or pGL3-P2 (*p* < 0.01). These results suggested that the promoter core region of the *MTNR1A* gene is located in the 540 bp region upstream of the TSS.

### 3.3. DNA Methylation Analysis of the Core Promoter in the MTNR1A Gene

To measure DNA methylation of the core promoter in the *MTNR1A* gene and to prepare samples for pyrosequencing, we split the core promoter of the *MTNR1A* gene into two fragments. The first fragment harbored 13 CpG sites, and the second segment harbored 14 CpG sites (Figure 3A). We performed pyrosequencing of these 27 CpG sites in the core promoter region. In the first fragment, the DNA methylation level of CpG site 1 in the LP42 group sample was significantly higher than that in the SP42 group sample (*p* < 0.01), and the methylation levels of CpG sites 4 and 8 in the LP42 group sample were higher than those in the SP42 group sample (*p* < 0.05) (Figure 3B). In addition, the DNA methylation level of two sites (7 and 8) in the SP-LP42 group sample was significantly higher than that in the SP42 group sample (*p* < 0.01) (Figure 3C). However, no significant difference in site methylation levels was found between each compared group in the second fragment (Figure 3D,E). These results indicate that the photoperiod can induce DNA methylation of the *MTNR1A* gene.

### 3.4. Correlation between DNA Methylation and MTNR1A Gene Expression

We statistically analyzed the differences between the LP42 vs. SP42 group sites and between the SP-LP42 and SP42 group sites to determine their significance, and we calculated the mean methylation level at the differential DNA methylation sites. Then, a correlation between the average methylation level of the differential DNA methylation site and the gene expression of MTNR1A was determined. The results of the correlation analysis revealed a significant negative correlation among photoperiod treatment groups (*p* < 0.001) (Figure 4), and the value of the Pearson correlation coefficient of the methylation level at the site with a significantly different methylation level and *MTNR1A* gene expression was −0.94 in the SP42 group, −0.964 in the LP42 group, and 0.963 in the SP-LP42 group.

## 4. Discussion

MLT is an indole hormone synthesized in the pineal gland, and its secretion is significantly regulated by photoperiod-induced signaling [23,24]. Melatonin receptor 1A is the most critical receptor in MLT release and plays an important regulatory role in mammalian reproductive functions, especially in the estrus of sheep [10,25,26]. Sheep are short-day breeders, and in addition to the influence of temperature, forage, and other environmental factors, the intrinsic change in rhythm caused by seasonal photoperiod is key to their reproduction. Environmental changes are often accompanied by epigenetic changes and exert a significant impact on animal reproduction, development, aging, etc. [27,28]. In our previous study, we constructed a bilateral ovariectomy model with Sunite sheep and analyzed the transcriptome in the hypothalamus after different artificial photoperiod treatments to study the regulatory mechanism of photoperiods on sheep seasonal reproduction. We found that the transcriptome level in the hypothalamus was significantly altered by photoperiod changes and that the expression of the *MTNR1A* gene was significantly higher in a short photoperiod than in a long photoperiod [13]. Therefore, we speculated that photoperiod may induce or otherwise contribute to DNA methylation, resulting in significant differences in gene or protein expression.

Previous studies have shown that polymorphisms in the *MTNR1A* gene are associated with reproductive seasonality [22,29], reproductive resumption [10], sexual activity of rams [30], and lamb birth weights [31] among sheep. In addition, *MTNR1A* is expressed in the hypothalamus, pituitary gland, and ovaries in several sheep breeds [10,11,32], and a significant difference in the expression of the *MTNR1A* gene in the hypothalamus between sheep exhibiting year-round estrus and sheep exhibiting seasonal estrus has been reported [22]. MLT plays a regulatory reproductive role with nocturnal secretion from the pineal gland that varies by photoperiod. The reception of neuroendocrine cell signaling triggered by photoperiod changes is mediated through MLT receptors. As the target of MLT, *MTNR1A* in the hypothalamus plays a vital role in photoperiod-induced reproductive regulation. In our study, we established a sheep model of estrus via ovariectomy, which has been previously performed for the functional study of the mammalian hypothalamus in rats, mice, Siberian hamsters, goats, and sheep [21,22]. In this model, an SP simulated estrus in the sheep under natural conditions, and an LP induced anestrus in the sheep. qPCR and Western blotting showed that the expression of MTNR1A in the hypothalamus at both the mRNA and protein levels in the SP42 sheep was significantly higher than that in the SP42 and SP-LP42 sheep (*p* < 0.05). These results suggested that the expression level of this gene during estrus is significantly higher than that in anestrus, which is completely consistent with the results of previous studies [11,33]. The results also indicated that the expression of MTNR1A in the hypothalamus is significantly regulated by photoperiod changes.

DNA methylation at the cytosine base in the 5th position has been associated with a plethora of biological roles in mammalian gene regulation, from cell differentiation to genomic imprinting and X-chromosome inactivation [34]. Many external factors in the natural environment (such as temperature and light) can modify genomic DNA through DNA methylation, thereby regulating the transcription of candidate genes and affecting production traits. A study showed that the postnatal exposure of mice to LP conditions induced an increase in TET2-dependent DNA hydroxymethylation in the hippocampus, a condition that might be involved in the long-term effects of the postnatal photoperiod on neurogenesis and affective/cognitive behaviors [35]. Studies have shown that DNA methylation regulates gene expression and plays an important role in the regulation of animal reproduction [36,37]. For example, the DNA methylation status of the *dio3* gene promoter region in the hypothalamus of Siberian hamsters was reversible with changes in the light cycle and was closely related to seasonal reproduction [19]. The promoter is a specific sequence that initiates gene transcription, and transcription factors bind to these regions to promote or inhibit transcription. Large-scale manipulation of promoter DNA methylation revealed context-specific transcription responses and transcript stability [34], and DNA methylation of CpG sites in a promoter can affect transcription factor binding, resulting in transcriptional repression and ultimately reducing gene expression [38,39]. Although the *MTNR1A* gene is thought to be related to ovine estrus or reproduction, it has not been studied from the perspective of DNA methylation in the promoter region. One previous study reported that the alteration of DNA methylation in the putative promoter region of the *MTNR1A* gene may induce pathways that increase cancer risk for nightshift workers [40]. In our present study, using a luciferase reporter gene, we constructed vectors carrying different lengths of the *MTNR1A* gene promoter sequence and identified the core promoter region to be in the 540 bp region upstream of the TSS (Figure 2). To meet sequencing requirements, we split the core promoter of the *MTNR1A* gene into two fragments. The first fragment carried 13 CpG sites, and the second fragment carried 14 CpG sites (Figure 3). The DNA methylation level of the 27 CpG sites was measured by pyrosequencing, and some CpG sites in the core promoter region showed significantly increased methylation under LP conditions, suggesting that photoperiod induces DNA methylation changes that affect the role played by MTNR1A in sheep reproduction. In addition, we analyzed the correlation between gene expression and the DNA methylation level, and in each sheep group, a significant negative correlation was identified (Figure 4). The results of this study are consistent with the general regulatory pattern of DNA methylation in gene transcription [41]. The above-mentioned results suggest that LP may induce an increase in the DNA methylation level at key CpG sites in the core promoter region of the *MTNR1A* gene, leading to inhibited gene transcription.

## 5. Conclusions

In summary, we located the core promoter region in the *MTNR1A* gene within a 540 bp upstream of the TSS. Pyrosequencing showed that the photoperiod induces DNA methylation at a cytosine site in the core promoter region of the *MTNR1A* gene. In addition, the level of DNA methylation was higher after LP, and there was a significant negative correlation with the expression of MTNR1A. These results suggested that the photoperiod may induce DNA methylation of the core promoter region in MTNR1A to regulate gene and protein expression, leading to changes in reproductive hormone secretion and contributing to the seasonal reproductive activities of ewes. We revealed the effects of the photoperiod on gene expression and sheep reproductive activity from the perspective of epigenetic changes, and these findings suggest new ideas for studying seasonal estrus and reproduction in sheep.

## Figures and Tables

**Figure 1 animals-13-01917-f001:**
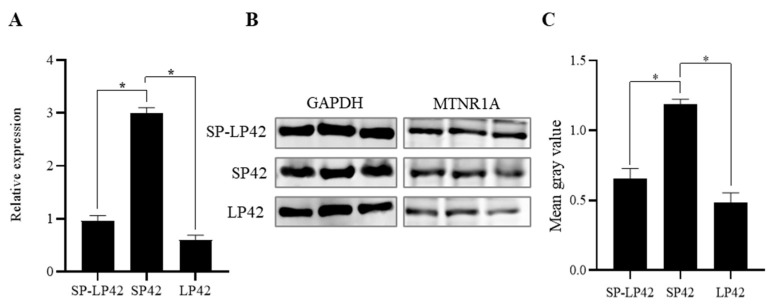
Expression of MTNR1A in the hypothalamus after different photoperiod treatments. (**A**) qRT-PCR analysis of the expression of MTNR1A in groups exposed to different photoperiods. (**B**) Western blots showing the expression of GAPDH and MTNR1A in different photoperiods. (**C**) Relative expression of MTNR1A according to a protein grayscale analysis. * *p* < 0.05.

**Figure 2 animals-13-01917-f002:**
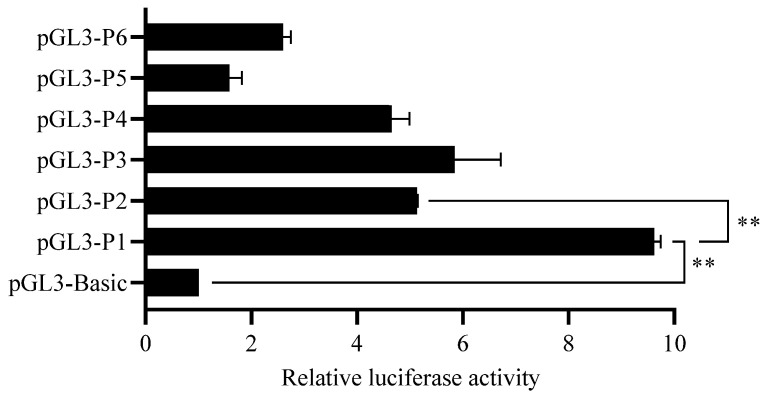
Luciferase assay. Deletion mutation constructs were transfected into HEK293T cells. Results are expressed as mean ± SEM (*n* = 3) in arbitrary units based on firefly luciferase activity level normalized to Renilla luciferase activity level. Results represent an average of three independent experiments performed in triplicate. ** *p* < 0.01.

**Figure 3 animals-13-01917-f003:**
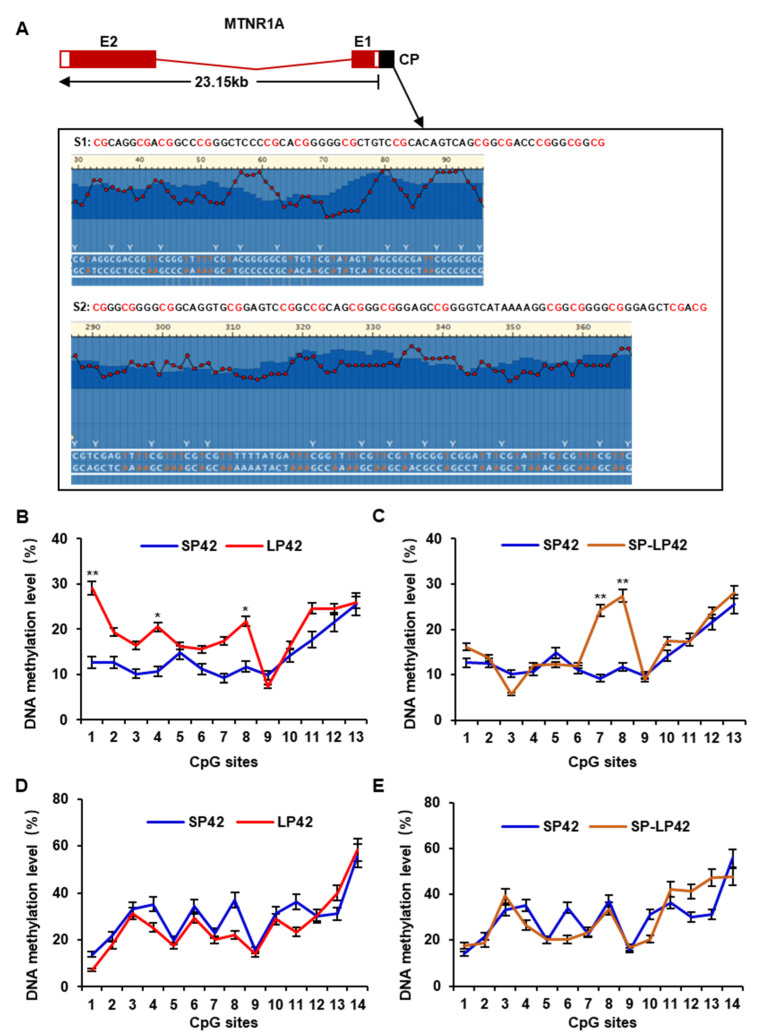
Methylation level of each CpG site in the core promoter region. (**A**) Sequence structure of the *MTNR1A* gene and CpG sites amplified in the core promoter region. E represents an exon, and CP represents the core promoter. S1 and S2 represent the core promoter sequences identified to be subsequently sequenced, where S1 carries 13 CpG sites, and S2 carries 14 CpG sites. (**B**–**E**) DNA methylation levels of 27 CpG sites under different photoperiod conditions. Significance is expressed based on comparisons with the SP42 group, * *p* < 0.05, ** *p* < 0.01. E1 and E2 represent exons 1 and 2, respectively, in the *MTNR1A* gene.

**Figure 4 animals-13-01917-f004:**
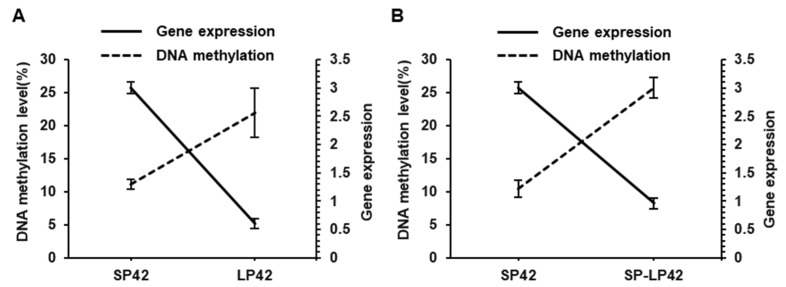
Relationship between MTNR1A expression and methylation level at each CpG site in the promoter region. (**A**) Relationship between methylation levels at sites with significantly different DNA methylation levels and gene expression between SP42 and LP42 groups. (**B**) Relationship between methylation levels at sites with significantly different DNA methylation levels and *MTNR1A* gene expression between SP42 and SP-LP42 groups.

**Table 1 animals-13-01917-t001:** Primer information for qPCR.

Primer Name	Sequences (5′-3′)	Product Size (bp)
MTNR1A-F	CCTCAGATACGGCAAGCTG	127
MTNR1A-R	GATCCTCGGGTCATACTGCA	
ACTB-F	GCTGTATTCCCCTCCATCGT	97
ACTB-R	GGATACCTCTCTTGCTCTGG	

**Table 2 animals-13-01917-t002:** Primer information for pyrosequencing.

Primer Name	Sequences (5′-3′)	Product Size (bp)
MTNR1A-F1	AAGAAGGAGTAGGGTGTTTTTG	274 bp
MTNR1A-R1	ACTACCCTTACCCTTAAAAATCCC	
MTNR1A-S1	CCCCCCCCCAAACACCTAA	
MTNR1A-F2	ATGTTTATTAAGATGGTGAAGATGAG	396 bp
MTNR1A-R2	TTAAAAAAAACCCAAAATACCCTTAAA	
MTNR1A-S2	GGTGGATTTTTAGAG	

## Data Availability

Not applicable.

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
