# Peer review of "Photoperiod Induces DNA Methylation Changes in the Melatonin Receptor 1A Gene in Ewes"

_animals, 2023, doi:10.3390/ani13121917_

Round 1

Reviewer 1 Report

This study investigates the existence of alterations in the DNA methylation level of the melatonin 1A receptor, MTNR1A, and its association with seasonal reproductive responses mediated by environmental photoperiod in ewes. The authors show that there are changes in the expression and protein levels of the MTNR1A gene in the hypothalamus of ovariectomized ewes that had been treated with estradiol, and subsequently subjected to different photoperiods: SP, LP, SP+LP. In a second phase, they proceeded to identify the promoter region of the MTNR1A gene and studied the level of DNA methylation, so that they found that under LP there was an increase in this methylation and that this correlated with a reduction in MTNR1A gene expression. Overall, the study seems well focused and offers novel results regarding the role of MTNR1A receptor in the seasonal regulation of reproduction in sheep.

The text of the manuscript has frequent punctuation mistakes and some grammatical expressions and writing errors that make it difficult to follow some paragraphs. As example, I cite the second paragraph of the introduction in which the text appears as a continuous whole, lacking punctuation and with very long sentences that make comprehension difficult. The Ms needs revision by English language experts.

It is necessary to provide more detailed information on the experimental procedures. At what time of day were the animals sampled? Was the sampling time the same in all three experimental groups?

How the SP-LP group was formed. How long did the ewes remain in SP before being transferred to LP? 

The reasons for using ovariectomized ewes that were subsequently treated with estradiol are not clear. There is a lack of information on the adequacy of this procedure to the objectives of the experimentation carried out.

Were the plasma estradiol levels monitored after treatments? And after subjecting the ewes to the different photoperiods? Why was a 42-day exposure to the different photoperiods (LP, SP, LP+SP) chosen?

On the other hand, in the text of point 3.3, reference is made to figure 4 (A-E) when in fact these results seem to correspond to figure 3. In relation to this figure, information should be included in the figure caption on the content of graphs B-E.

It seems a bit conflicting that ewes subjected to long photoperiod (LP) and displaying low MTNR1A gene expression and enzyme protein levels, compared to those kept in SP or SP-LP, exhibit a higher DNA methylation level of the MTNR1A promoter region. Perhaps the authors should discuss in more detail the inverse correlation between DNA methylation of the MTNR1A gene promoter region and MTN1A gene expression.

The text of the manuscript has frequent punctuation mistakes and some grammatical expressions and writing errors that make it difficult to follow some paragraphs

Reviewer 2 Report

In the present work, He et al. try to explain that photoperiod induces DNA methylation change of melatonin receptor 1A gene in ewes. It is interesting, but some questions also should be explained.

1. It only focuses on melatonin receptor 1A, how is about melatonin receptor 1B.

2. Editing of English language and style is very needed. The writing ability of authors is poor. The manuscript should be revised throughout by a SCIENTIFIC expert. Pease revise throughout this manuscript.

For example, Lines 12-13, ‘Accurate timing of photoperiod changes is a vital ability for animal reproduction.’

3. Lines 13, 25, please explain ‘MTNR1A’.

4. Line 17, 42days.

5. Lines 20, 34, ‘found there was’.

6. Lines 21, expression level of MTNR1A (italic), and many genes should be in italic throughout this manuscript.

7. Lines 32-33, please explain ‘the SP42 with the LP42 or SP-LP42’.

8. Line 37, ‘correlation(p<0.001)’.

9. Line 67, ‘[13]. a strong relationship’.

10. Line 73, studies (References 15, 16 and 17) about methylated genes are only performed in China?

11. Hypothesis and aim should be added in Introduction section.

12. Lines 100-102, ‘bilateral ovariectomy and estradiol treatments’. ‘In brief, Estradiol treatment was achieved with an inner diameter of 3.35 mm and out the diameter of 4.65 mm, packed with 20mg crystalline 17β-estradiol’. It is difficult to know the meaning. ‘bilateral ovariectomy’ not only results to no estradiol secreted from ovaries, but also no P4 and other hormones are secreted from ovaries. However, there is relationship between hypothalamus tissues and pituitary body. What is ‘an inner diameter of 3.35 mm’?

13. Line 104, please explain ‘E2’.

14. Line 120, ‘4% SurePAGE gel’? target protein molecular weight?

15. Lines 124-125, detail information for GAPDH antibody should be added.

Table 1. there is only one housekeeping gene (ACTB) used for PCR. GenBank accession no. should be added.

16. Line142, 293 at beginning of a sentence is not suitable.

17. Line 143, ‘CO2.’

18. Line 162, ‘ul’

19. Line 193, ‘LP42(p<0.05)’

20. Figure 2. is there no *p < 0.05?

21. Line 253, ‘Melatonin (MLT)’, abbreviation is only present in here, and not necessary.

22. Format of references is not suitable for this Journal. Paper titles only first letter of first word is in capitalization. Authors are not in a manner.

23. Line 359, ‘Fish. Reproduction.’

The writing ability of authors is poor. The manuscript should be revised throughout by a SCIENTIFIC expert. Pease revise throughout this manuscript.

Reviewer 3 Report

This research provides interesting information. However, some important changes need to be made before final publication.

Abstract: review the "Journal" guidelines. It is mentioned in "MDPI Style Guide" the following: "The abstract contains a summary of the entire paper and can be up to 200 words long with only one paragraph". (https://www.mdpi.com/authors/layout) In this case it exceeds the number of words. Therefore, restructure this section.

INTRODUCTION

General comments:

I recommend being more specific with the objective of this study.

Specific comments:

Line 62.- I recommend expanding the explanation on the "epigenetic" factors that modify gene expression regulation of melatonin.

Line 62.- revise the spelling "gene expression [13]. a strong relationship".

 MATERIAL AND METHODS

General comments: I recommend mentioning the type of production system and what were the selection criteria for these animals.  It would be convenient to mention during which period the study was carried out. In addition, it is necessary and important to mention the ethical and animal welfare criteria that were used for the management of these animals. Is there an ethics committee that evaluated this research?

Line 182.- it would be convenient to mention if the data came from a population with normal distribution or if they were transformed. How were the data represented? Are the assumptions of normality and homoscedasticity met? Because it is necessary for a One-Way or Two-Way ANOVA to be performed. Also, if the data did not have a normal distribution, a Spearman's correlation is recommended. Also, mention the type of significance used (P<0.05 or P<0.01).

DISCUSSION

In general, I recommend orienting the discussion according to how you reported your results.

I recommend being more specific with the comparison and explanation of your results with the discussion section. As well as expanding this section.

To elaborate more on "DNA methylation".

Clarify how is the mechanism of "MTNR1A".

Round 2

Reviewer 2 Report

Thanks for author’s responses. However, GenBank ID should been added.

Author Response

 GenBank ID should been added.

Response: Thank you very much for your carefulness and comments. We have added the GenBank ID in the manuscript.